

# Depth effect on the prokaryotic community assemblage associated with sponges from different rocky reefs

Bárbara González-Acosta[1],*, Aarón Barraza[2],*,
César Guadarrama-Analco[1], Claudia J. Hernández-Guerrero[1],
Sergio Francisco Martínez-Díaz[1], César Salvador Cardona-Félix[3] and
Ruth Noemí Aguila-Ramírez[1]

[1] Instituto Politécnico Nacional-Centro Interdisciplinario de Ciencias Marinas, La Paz, Baja California Sur, México
[2] CONACYT-Centro de Investigaciones Biológicas del Noroeste, La Paz, Baja California Sur, México
[3] CONACYT-Instituto Politécnico Nacional, La Paz, Baja California Sur, México
* These authors contributed equally to this work.

Corresponding author
Ruth Noemí Aguila-Ramírez,
raguilar@ipn.mx

## ABSTRACT

**Background:** Sponge microbiomes are essential for the function and survival of their host and produce biologically active metabolites, therefore, they are ideal candidates for ecological, pharmacologic and clinical research. Next-generation sequencing (NGS) has revealed that many factors, including the environment and host, determine the composition and structure of these symbiotic communities but the controls of this variation are not well described. This study assessed the microbial communities associated with two marine sponges of the genera *Aplysina* (Nardo, 1834) and *Ircinia* (Nardo, 1833) in rocky reefs from Punta Arena de la Ventana (Gulf of California) and Pichilingue (La Paz Bay) in the coast of Baja California Sur, México to determine the relative importance of environment and host in structuring the microbiome of sponges.

**Methods:** Specimens of *Aplysina* sp were collected by scuba diving at 10 m and 2 m; *Ircinia* sp samples were collected at 2 m. DNA of sponge-associated prokaryotes was extracted from 1 cm$^3$ of tissue, purified and sent for 16S amplicon sequencing. Primer trimmed pair-ended microbial 16S rDNA gene sequences were merged using Ribosomal Database Project (RDP) Paired-end Reads Assembler. Chao1, Shannon and Simpson (alpha) biodiversity indices were estimated, as well permutational analysis of variance (PERMANOVA), and Bray-Curtis distances.

**Results:** The most abundant phyla differed between hosts. Those phyla were: Proteobacteria, Acidobacteria, Cyanobacteria, Chloroflexi, Actinobacteria, Bacteroidetes, and Planctomycetes. In *Ircinia* sp the dominant phylum was Acidobacteria. Depth was the main factor influencing the microbial community, as analysis of similarities (ANOSIM) showed a significant difference between the microbial communities from different depths.

**Conclusion:** Microbial diversity analysis showed that depth was more important than host in structuring the *Aplysina* sp and *Ircinia* sp microbiome. This observation contrast with previous reports that the sponge microbiome is highly host specific.

# INTRODUCTION

Marine sponges (MS) inhabit shallow to mesophotic ecosystems and harbor on diverse symbionts (*Taylor et al., 2007*; *Simister et al., 2012*) that reach up to 50% of their total weight (*Hentschel et al., 2003*; *Usher et al., 2004*). Frequent and abundant presence of bacteria, especially within the sponge mesohyll, led authors to address these bacteria as symbionts (*Vacelet, 1975*; *De Vos et al., 1995*; *Burja et al., 1999*; *Imhoff & Stöhr, 2003*). Sponge microbiomes are essential for their host's function (metabolic), health and survival (*Lurgi et al., 2019*). Furthermore, production of biologically active metabolites by sponges-associated bacteria, is an important function in this association (*Imhoff & Stöhr, 2003*). Therefore, they are ideal candidates for ecological, pharmacological and clinical research. Sponge tissues host many symbionts, including heterotrophic bacteria, facultative anaerobes, dinoflagellates, cyanobacteria, archaea, fungi, and viruses (*Webster & Hill, 2001*; *Schippers et al., 2012*). These microbial communities included dozens of phyla. Variation in diversity of these microbiomes is not well described (*Taylor et al., 2007*; *Webster & Thomas, 2016*; *Villegas-Plazas et al., 2019*).

Next-generation sequencing (NGS) approaches to characterize marine sponge microbial communities has dramatically increased precision and quantity of surveys of the taxonomic complexity of these microbiomes (*Schmitt et al., 2011*; *Webster & Taylor, 2012*; *Reveillaud et al., 2014*) and revealed that sponge microbiomes are largely host-specific and stable across temporal scales under specific environmental conditions (*Morrow et al., 2015*; *Weigel & Erwin, 2015*; *Morrow, Fiore & Lesser, 2016*; *Cleary et al., 2019*). Recent research suggests that depth drivers of the structure of ocean microbiome (*Sunagawa et al., 2015*). However, for symbioses, one would expect a strong microbial community differentiation to emerge across host species (*Lurgi et al., 2019*). Sponges can inhabit from shallow to mesophotic ecosystems, in deep water they are apparently less influenced by abiotic factors (*Kahng, Copus & Wagner, 2014*; *Olson & Kellogg, 2010*). In shallow water these abiotic factors could influence the sponges, and their associated microbial communities. Some studies have determined sponge associated microbial community changes at different water depths from shallow (0–30 m) to mesophotic areas (30–150 m) (*Olson & Kellogg, 2010*; *Lesser, Slattery & Leichter, 2009*; *Kahng, Copus & Wagner, 2014*). Though the specificity of the sponge microbiota appears more related with host phylogeny, microbial communities in shallow and deep reefs vary (*Steinert et al., 2016*)

However, to our knowledge no studies are available that evaluate whether among the same shallow water sponges (0–30 m) the community varies according to its range of distribution. Although changes in abiotic factors are not as evident, as it could occur in mesophotic zones (30–150 m), the depth gradient could influence the composition of the microbial community associated with these sponges (*Olson & Gao, 2013*; *Steinert et al., 2016*).

*Aplysina* species are often associated with shallow rocky reefs. This species belongs to the Verongiida order and are distributed along the East Pacific from Mexico to Panama

(*Caballero-George et al., 2010*; *Cruz-Barraza et al., 2012*). *Ircinia*, they are conspicuous and abundant in areas exposed to light in rocky-coral biotopes (*Parra-Velandia & Zea, 2003*) and more abundant in localities near sources of continental discharge with greater turbidity and load of organic material in suspension (*Zea, 1994*). In previous studies with sponges of these genera (*Aplysina* sp and *Ircinia* sp) from the Gulf of California, differences were observed in the biological activity of sponges and their associated bacteria, between sponges of the same genus and between genera (*Gándara-Zamudio, 2011*; *Aguila-Ramírez, 2012*; *Ortíz-Aguirre, 2012*). These differences appear related to site and depth. For this reason, this study assessed the microbial communities associated with two marine sponges of the genera *Aplysina* (Nardo, 1834) and *Ircinia* (Nardo, 1833) in rocky reefs from Punta Arena de la Ventana (Gulf of California) and Pichilingue (La Paz Bay) in the coast of Baja California Sur, México to determine the relative importance of environment and host in structuring the microbiome of sponges.

## MATERIALS AND METHODS

Specimens of *Aplysina* sp and *Ircinia* sp sponges previously collected to evaluate their biological activity (*Gándara-Zamudio, 2011*; *Aguila-Ramírez, 2012*; *Ortíz-Aguirre, 2012*) were used for this study.

*Aplysina* sp specimens ($n = 8$) were collected in triplicate by scuba diving in Punta Arena, Baja California Sur, México (24°03′40″N and 109°49′52″W) at different water depths (2–10 m). For this study, the depth of 2 m was considered shallow and 10 m as deep (Apl-S: 2 m; Apl-D: 10 m).

Three specimens in triplicate of *Ircinia* sp were collected in the Pichilingue locality inside La Paz Bay in Baja California Sur (24°16′08″N and 110°19′39″W) at 2 m depth (Fig. 1) (Permit SEMARNAT-08-049b Positive Ficta). Sponge samples were placed in sterile plastic bags and transferred to ice. In the laboratory, the epibiont organisms were removed and washed three times with sterile natural sea water, the outermost layer or pinacoderm was separated with a scalpel and pieces were cut from different areas of the sponges according to the suggestion by *Friedrich et al. (2001)*, placed in tubes and frozen at −20 °C. Dr. Cristina Vega Juárez from the Bentos Laboratory of the Institute of Marine Sciences and Linmology of the Universidad Autónoma de México (UNAM), identified the sponges using dichotomous keys and published bibliography for the East Pacific on sponge taxonomy (*Gómez et al., 2002*; *Cruz-Barraza & Carballo, 2008*; *Carballo-Cenizo & Cruz-Barraza, 2010*).

### Total DNA extraction

Pieces of approximately 1 cm$^3$ of sponge mesohyl were taken from each of sample and their replicates to form a composite sample; composites were then finely fragmented with a scalpel; 500 µL of TE buffer (10 mM Tris-HC1, 1 mM disodium ethylenediaminetetraacetic acid (EDTA), Thermo Fisher Scientific, Waltham, MA, USA) were added and sonicated for 10 min to detach the bacteria (Bransonic 3510). Then, the mixture was centrifuged at 8,000× g for 10 min, and the supernatant was placed in another 2 ml tube for DNA extraction with phenol: chloroform: isoamyl following *Sambrook & Russell (2002)* and
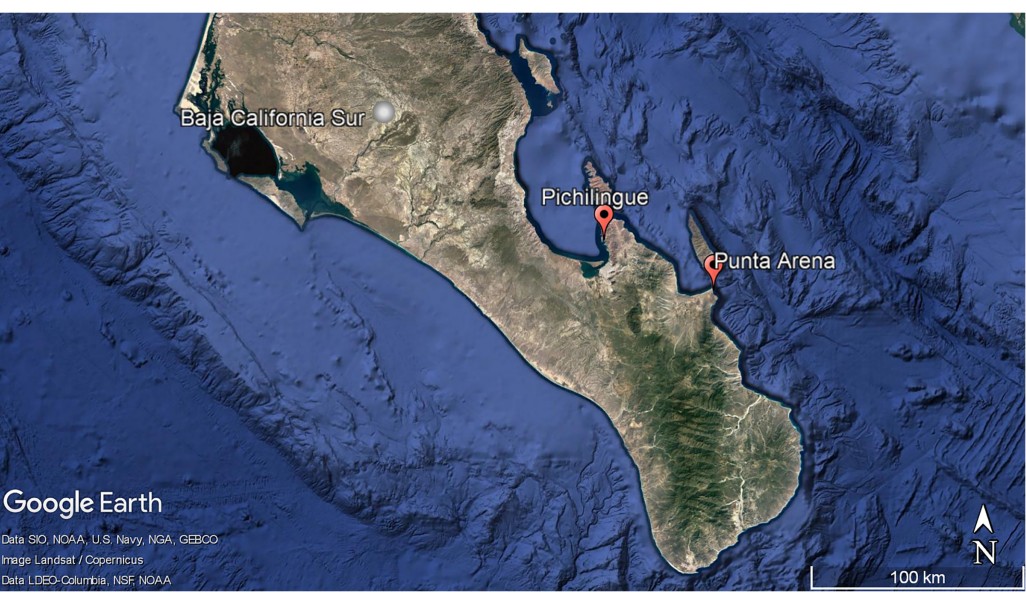

**Figure 1 Geographic localization of sponge sample collecting. Pichilingue and Punta Arena locations in Baja California Sur.** Each station is indicated in red. White bar represents 100 km. Map data © Google, Data SIO, NOAA, U.S. Navy, NGA, GEBCO, Image Landsat/Copernicus. Data LDEO-Columbia, NFS, NOAA

*Caamal-Chan et al. (2019).* Recovered DNA was resuspended in 50 µl of TE buffer (pH 8.0) and treated with RNase A (10 mg mL$^{-1}$, Promega, Madison, WI, USA) at 37 °C for 30 min. The integrity of the DNA was analyzed by agarose gel electrophoresis. Purity (λ 260 nm/280 nm ratio) and quantity were evaluated with a NanoDrop 2000 spectrophotometer (Thermo Fisher Scientific, Waltham, MA, USA). DNA samples were stored at −20 °C.

## 16S V4 rDNA sequencing

Purified DNA was sent to the Next Generation Sequencing Core at Argonne National Laboratory, Argonne, IL, USA for amplicon sequencing. Briefly, the microbial 16S rRNA gene V4 regions were amplified using primer set 515F (5′-GTGC CAGCMGCCGCGG TAA-3′) and 806R (5′-GGAC TACHVGGG TWTCTAAT-3′) following the method described by *Kozich et al. (2013)*. Amplicons of 16S rRNA gene V4 regions were generated using Illumina MiSeq 500-cycle kit with Illumina MiSeq sequencing platform (San Diego, CA, USA).

## Sequence processing and microbial diversity analysis

Primer trimmed pair-end bacterial 16S rDNA gene sequences were merged using Ribosomal Database Project (RDP) Pair-end Reads Assembler. The assembled sequences with an expected maximum error adjusted Q score less than 25 over the entire sequence were eliminated. VSEARCH (v2.4.3, 64 bit) was used to remove chimeras *de novo*, followed by removing chimeras by reference using RDP 16S rDNA gene (*Rognes et al., 2016*). High quality and chimera-free sequences were then clustered at 97% sequence similarity by CD-HIT (4.6.1) and RPD Classifier with a confidence cutoff at 50% (*Cole et al., 2014*).

These sequences resulted in the identification of unique operational taxonomic units (OTUs) and their abundance in each sample (*Wang et al., 2007*; *Fu et al., 2012*; *Bonder et al., 2012*; *Chen et al., 2013*). The resulting operational taxonomic unit (OTU) table was then processed to be analyzed with R programming language, using various packages and custom scripts (www.r-project.org). Chao1 and Shannon and Simpson (alpha) biodiversity indices were estimated with the package 'iNEXT' (*Hsieh, Ma & Chao, 2016*). For data normalization, the frequency of best hits to each individual taxon for each metagenome was divided by the total number of hits per sample. PERMANOVA and ANOSIM statistical analysis were performed with the 'adonis' and 'anosim' functions, respectively, with the package 'vegan'. Bray-Curtis distance estimations were calculated using the 'vegdist' function, as well as principal coordinate analysis using the 'pcoa' function with the package 'vegan' (*Oksanen et al., 2014*).

## RESULTS

### Sequencing run metrics

From all the samples sequenced, 379,392 reads were generated; after processing, 85,818 low-quality reads and chimeras were removed to keep high-quality pair-end-assembled reads, of which 146,787 reads could be assigned to prokaryotic taxa. The sequencing effort was assessed by Good's coverage analysis with a mean value for all sample reads of 71% ± 0.1% and a completeness analysis (full coverage reached below 5,000 reads) (Fig. S1). A total of 1,102 OTUs were obtained by similarity clustering at 99% nucleotide identity and 786 OTUs after singleton removal within each sample. Raw sequence reads are deposited in NCBI BioProject Database Accession number: PRJNA760541.

### Microbial communities associated with sponges

The resulting OTUs for this study showed that seven phyla were the most abundant among all mesohyl samples from *Aplysina* sp (10 m depth), *Aplysina* sp and *Ircinia* sp (2 m depth) with a minimal presence of Archaea (0.56–1.65% of the total classified reads). Proteobacteria represented the most abundant phylum (86%) for *Aplysina* sp (10 m) and was among the most abundant for *Aplysina* sp and *Ircinia* sp (2 m) samples (31% for both) (Fig. 2A).

Acidobacteria represented the most abundant (47%) phylum for *Ircinia* sp samples and also the most abundant (9%) for *Aplysina* sp (2 m) sample (Fig. 2A). Cyanobacteria was among the most abundant (5–22%) phylum for all samples, which was the second most abundant (22%) for *Aplysina* sp (2 m) sample (Fig. 2A). Chloroflexi was among the most abundant (16%) phylum for *Aplysina* sp at shallow water depth (Fig. 2A). Actinobacteria showed a higher abundance of 2- and 3-fold than *Aplysina* sp (10 m) for *Aplysina* sp and *Ircinia* sp samples at shallow depth (Fig. 2A). Bacteroidetes was among the most abundant phylum for *Aplysina* sp (5%) and *Ircinia* sp samples (Fig. 2A). Planctomycetes was among the most abundant phylum for shallow *Aplysina* sp (5%) samples (Fig. 2A). Interestingly, 30% of the total OTUs were shared among all samples analyzed (Fig. 2B). *Aplysina* sp (2 m) samples showed the highest amount of specific OTUs

a)

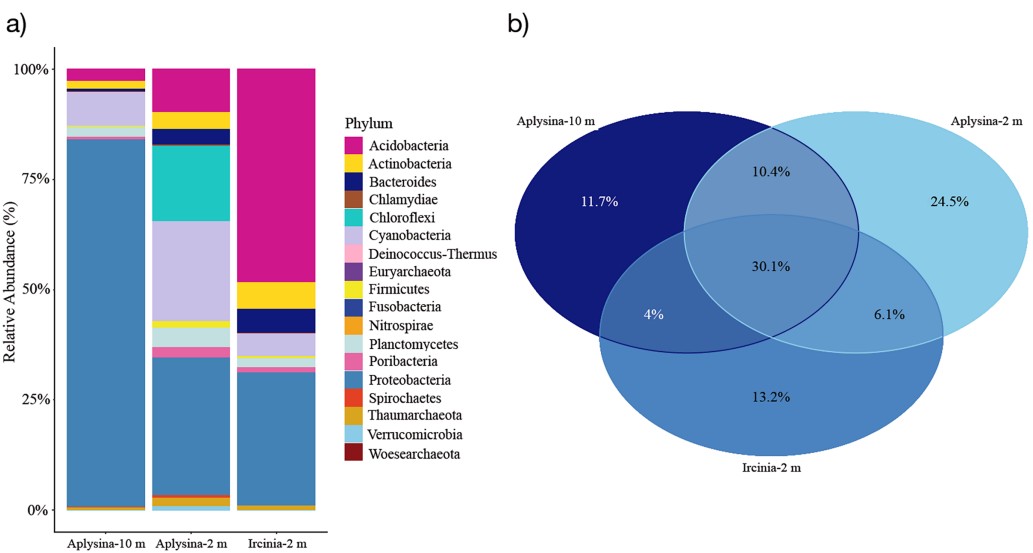

b)

**Figure 2** **Microbial assemblage of *Aplysina* sp and *Ircinia* sp at different depths.** (A) Microbial diversity structure and (B) Venn diagram for OTUs shared between *Aplysina* sp and *Ircinia* sp samples at different depths.

(25%), followed by *Ircinia* sp 13% and *Aplysina* sp (10 m) 12% samples (Fig. 2B). Both *Aplysina* sp samples (2 and 10 m) showed the highest ratio of exclusively shared OTUs (10%) (Fig. 2B).

## Microbial community diversity and depth effect on sponge species

Sampling curve analysis for Chao1 (order q = 0) (Fig. 3B), Shannon (order q = 1) (Fig. 3C), and Simpson (order q = 2) (Fig. 3D) indices showed differences due to curve clustering of the samples analyzed (Fig. 3). Furthermore, the principal coordinates analysis (PCoA) and constrained correspondence analysis (CCA) (Figs. 4A and 4B, respectively) showed two well-defined and discrete groups based on sample depth regardless of the species (*Aplysina* sp or *Ircinia* sp).

PERMANOVA analysis showed that depth was the main factor influencing the microbial community structures in the sponge samples ($R_2$ = 0.507, $P$ = 0.008). Sponge species did not have a significant effect ($R_2$ = 0.184, $P$ = 0.122). Moreover, the PERMANOVA analysis for depth interaction species showed that depth ($R_2$ = 0.507, $P$ = 0.004) was the main factor influencing microbial community structures regardless of sponge species ($R_2$ = 0.110, $P$ = 0.110). ANOSIM analysis also showed a significant difference between depths in the microbial communities (Fig. 5) and unsupervised bi-clustering analysis based on sample correlation to estimate the degree of relationship among samples supported this beta diversity analysis (PCoA and CCA) (Figs. S2, S3).

## DISCUSSION

This study characterized for the first time prokaryotic communities associated with *Aplysina* sp and *Ircinia* sp mesohyl sponges from the Gulf of California. These sponges were initially collected to evaluate their potential production of bioactive compounds and
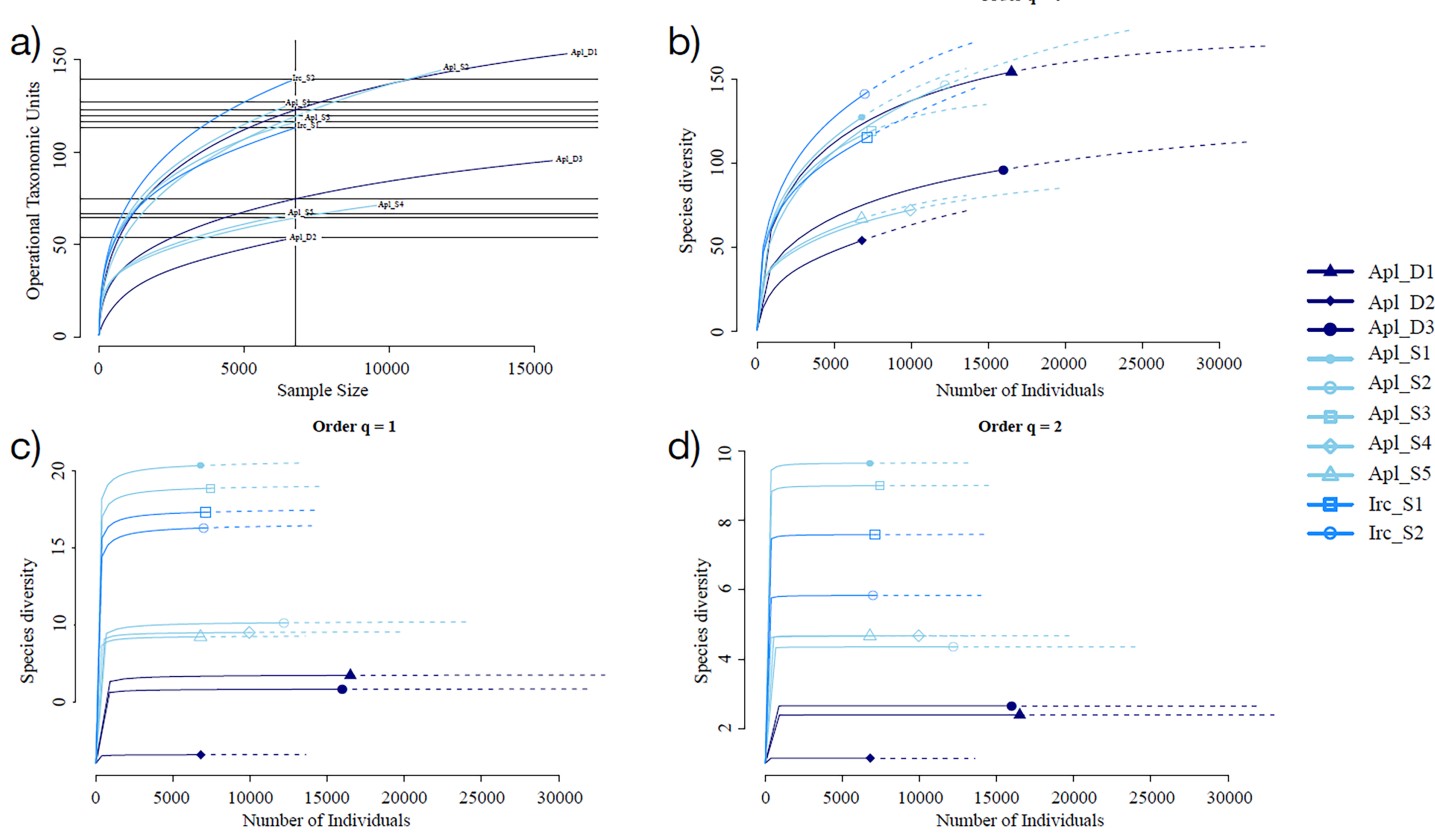

**Figure 3 Rarefaction sampling curves and alpha diversity estimations for *Aplysina* sp and *Ircinia* sp samples at different depths.** (A) Rarefaction sampling curves, (B) Chao1 index estimations ($q = 0$), (C) Shannon index estimations ($q = 1$), (D) Simpson index estimations through rarefaction (interpolation) and extrapolation (R/E) sampling curves.

to isolate the bacteria associated with them. Differences were observed in the biological activity of these sponges and cultivable bacteria isolated between the sponges of the same genera and between genera (*Gándara-Zamudio, 2011*; *Aguila-Ramírez, 2012*; *Ortíz-Aguirre, 2012*) appeared related to the site and depth at which they were collected, for this reason the prokaryotic communities from both sponges was characterized by high-throughput sequencing the 16S rRNA gene fragments.

*Ircinia* sp is only found in shallow areas without a depth slope; *Aplysina* sp, is distributed in a rocky reef that ranges from 2 to 10 m deep. Preliminary analysis the results of the sequencing, showed similarity between bacterial communities of sponges collected in the shallowest areas, therefore, the analysis was carried out focusing on the effect of depth on the bacterial diversity of these two genera of sponges.

Variation was found in relative abundance of the bacterial phyla associated with *Aplysina* sp at different depths. Archaea were present in a low abundance percentage. This low abundance could have been since specific primers for archaea were not used. *Chaib De Mares et al. (2017)* reported that when bacterial-specific primers were used, only 6% of the readings were classified as Archaea. On the other hand, when Archaea-specific primers were used, this proportion was 89%. The phylum Thaumarchaeota was the most abundant

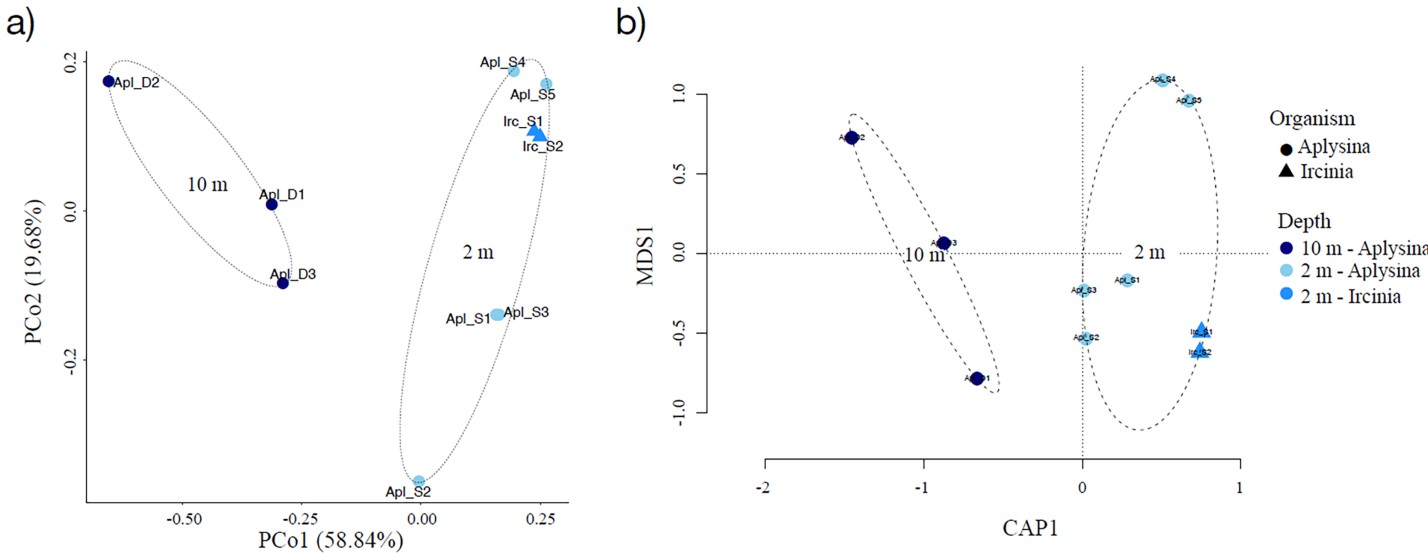

**Figure 4 Sample clustering analysis of *Aplysina* sp and *Ircinia* sp samples at different depths.** (A) Principal coordinates analysis and (B) canonical correspondence analysis for *Aplysina* sp and *Ircinia* sp samples at different depths.

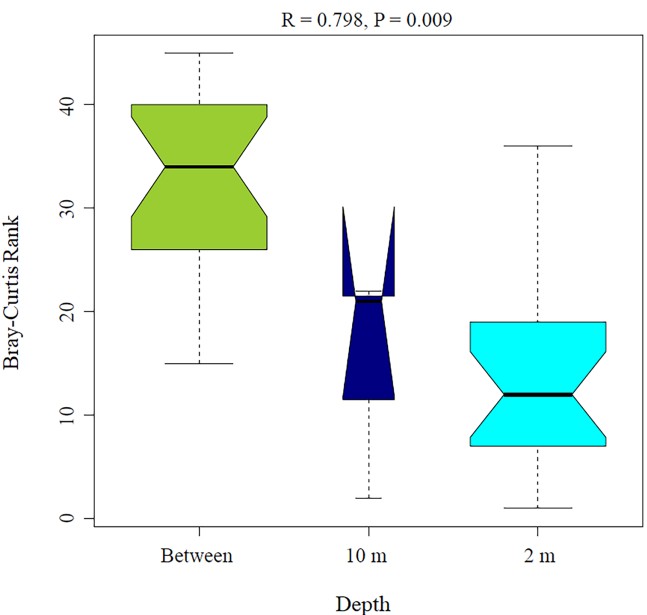

**Figure 5 Analysis of similitude (ANOSIM) between sponge samples.** ANOSIM analysis showed differences between *Aplysina* sp at 2m depth and *Aplysina* sp and *Ircinia* sp at 10 m depth samples.

within archaea and was found in all samples and was most abundant in *Aplysina* sp samples collected at 2 m. This phylum–Thaumarchaeota–comprises nitrifying archaea and is highly abundant in sponge microbiomes (*Dat et al., 2018*).

The most abundant phylum at both depths were Proteobacteria (Fig. 2A). Proteobacteria are a prominent group of sponge-associated microbial communities and highly abundant in the marine in plankton organisms (*Li et al., 2006*; *Jasmin et al., 2015*;

*Dat et al., 2018*). This phylum contributes to biogeochemical cycles through extracellular enzyme production and performs some symbiotic functions in sponges, such as nitrogen fixation and secondary metabolite production for the chemical defense of the host (*Stabili et al., 2014*). Furthermore, Cyanobacteria and Chloroflexi showed a high proportional abundance in the *Aplysina* sp samples from the shallow zone. These phyla mediate photosynthesis and carbon fixation and responsible for converting ammonia into nitrate in marine sponges (*Han, Li & Zhang, 2013*; *Bibi & Azhar, 2021*). The high proportion of these bacteria in the shallowest sponges could be explained because light intensity (*Erwin et al., 2012*; *Souza et al., 2017*; *Glasl et al., 2019*; *Fiore et al., 2020*).

As reported by *Hardoim et al. (2021)*, prokaryotic communities associated with *A. caissara* and *A. fulva* were very similar among these two species and dominated by Chloroflexi, Proteobacteria, Crenarchaeota, and Acidobacteria. In contrast, in this study, Chloroflexi was only represented with a high relative abundance in 2-m *Aplysina* sp samples and Crenarchaeota did not represent an important component in any of the samples. However, these same authors (*Hardoim et al., 2021*) mentioned that community composition was largely different for *Aplysina* species. For instance, the most abundant phyla encountered in *A. fulva, A. cauliformis, A. archeri, A. cavenicola*, and *A. aerophoba* sampled in seven distinct sites were assigned to Proteobacteria, Chloroflexi, unclassified bacteria, Acidobacteria, and Actinobacteria (*Thomas et al., 2016*), or *A. fulva* collected in Brazil with the community dominated by Cyanobacteria, Proteobacteria, and Chloroflexi (*Hardoim et al., 2009*), which coincides with the phyla with the highest relative abundance in the 2-m *Aplysina* sp samples collected at Punta Arena BCS.

Global bacterial community structure for shallow samples has a similar proportion of Proteobacteria phylum (Fig. 2A). For *Ircinia* sp samples Acidobacteria and Proteobacteria were the most abundant phyla, which is consistent with the results of several studies (*Mohamed, 2007*, *Schmitt et al., 2007*; *Schmitt et al., 2008*; *Lee et al., 2011*; *Yang et al., 2011*; *Hardoim et al., 2012*; *Pita, López-Legentil & Erwin, 2013*; *Pita Galán, 2014*; *Engelberts et al., 2020*) where they report that the core bacterial community associated with this genus is made up of seven phyla Proteobacteria, Acidobacteria, Cyanobacteria, Bacteroidetes, Actinobacteria, Firmicutes and Nitrospira. Despite being a very abundant and diverse group, the Acidobacteria phylum is not as well studied as Proteobacteria, so very little information is available on the species belonging to this phylum in marine environments. *Engelberts et al. (2020)* analyzed specific genes involved in metabolic pathways and biogeochemical cycles and found that some species of Acidobacteria participate in denitrification, nitrification, ammonification, metabolism of the taurine, exopolysaccharide production and synthesis of B complex vitamins. This phylum has also been found in a high percentage in sponge species, such as *Xestospongia testudinaria* and *Luffariella variabilis*, but these bacteria had not been reported as predominant in sponges of the genus *Ircinia* (*Webster et al., 2013*). In most studies of the community associated with different species of *Ircinia*, Proteobacteria and Cyanobacteria dominate (*Hardoim & Costa, 2014*). It should be noted that this study would be the first report where the abundance of Acidobacteria is greater than that of Proteobacteria for the genus *Ircinia*. For the shallow *Aplysina* sp samples, the most

abundant were Proteobacteria, Cyanobacteria, Chloroflexi and Acidobacteria; deep *Aplysina* sp samples were dominated only by the Proteobacteria phylum. These prokaryotic taxa with a high relative abundance in this study are also abundant in other marine sponges (*Moitinho-Silva et al., 2017*; *Dat et al., 2018*).

In coral ecosystems host identity controls the sponge-associated microbial community (*Steinert et al., 2016*). This study found that sponges of different species (*Aplysina* sp and *Ircinia* sp) collected at different depths (2 and 10 m) share a high proportion of OTUs (Fig. 2B). Albeit the *Aplysina* sp samples at 10 m and those at 2 m showed characteristic and discrete OTUs distributions (Fig. S3). Environmental variability is an important factor in the sponge microbial community. Stable isotopic analysis in giant barrel sponge *Xestospongia muta* showed changes in the relationship $_{15}N/_{13}C$ in sponges as depth increased (transition from dependency on photoautotrophy to heterotrophy), leading to a more stable microbial community along the depth gradient (*Morrow, Fiore & Lesser, 2016*).

A high proportion (~30%) of shared OTUs for *Aplysina* sp and *Ircinia* sp might be due to the ecophysiological similitudes among those species, since they are inhabiting similar reef areas and face similar selective pressures (*Souza et al., 2017*; *Pearman et al., 2019*; *Turon et al., 2019*). However, a higher proportion of exclusively shared OTUs for *Aplysina* sp samples was expected for both depths (10%) since those samples belong to the same species, and because other studies have found that host-specific prokaryotic communities are stable despite geographical and temporal differences (*Erwin et al., 2015*; *Hardoim & Costa, 2014*; *Dat et al., 2018*). Approximately the same proportion of exclusively shared OTUs are shared between *Aplysina* sp at shallow and deep depths, respectively. Several phyla are stable in *Aplysina* sp samples collected at different depths, but their relative abundance percentage differs markedly. In the deepest samples a clear Proteobacteria predominated, while in the shallowest samples three phyla, Proteobacteria, Cyanobacteria and Chloroflexi, predominated. Although this study does not have data on environmental parameters, other studies have found that the temperature difference between shallow areas of the reef and deep sites averaged 4 °C (from 3 to 91 m deep), which was unlikely to affect sponge-microbial communities. Studies examining the effect of elevated temperatures found no change (at sub-lethal temperatures) in sponge bacterial communities during short term experiments (*Webster et al., 2008*; *Simister et al., 2012*; *Steinert et al., 2016*). Therefore, and due to the presence of bacteria of the phylum Cyanobacteria and Chloroflexi in greater abundance at 2 m, this might imply that light intensity plays an important role in community changes, as other authors have suggested (*Lesser et al., 2010*).

Alpha diversity indices for richness and evenness estimated with rarefaction (interpolation) and extrapolation (R/E) sampling curves showed the clustering for *Aplysina* sp deep samples and overlapping both *Aplysina* sp and *Ircinia* sp shallow samples (Fig. 3). The beta diversity PCoA and CCA analyses showed a clustering directly related with depth instead of a relationship among sponge species supported with PERMANOVA and ANOSIM analyses (Figs. 3 and 4). The clustering analyses (PCoA and CCA) showed two well-defined groups, one corresponding to *Aplysina* sp bacteria

collected in deeper areas and the other one corresponding to *Aplysina* sp and *Ircinia* sp bacteria from the shallow area. These findings are in accordance with the variation in the bacterial community structure assemblage, which is influenced directly by environmental factors, such as depth, temperature, and light intensity and not by the host sponge species (*Maldonado & Young, 1998*; *Thoms et al., 2003*; *Olson & Gao, 2013*; *Morrow, Fiore & Lesser, 2016*; *Thomas et al., 2016*; *Pearman et al., 2019*; *Souza et al., 2017*). These results differ from those reported in other studies. For example, *Gantt et al. (2019)* reported that bacterial communities exhibited a high degree of host specificity.

## CONCLUSIONS

Microbial diversity analysis showed that depth was more important than host in structuring the *Aplysina* sp and *Ircinia* sp microbiome.

## ACKNOWLEDGEMENTS

The authors thank José Borges Souza for his help in taking samples, Cristina Vega Juárez by identifying the sponges, SNI (Sistema Nacional de Investigadores) COFAA (Comisión de Operación y Fomento de Actividades Académicas del Instituto Politécnico Nacional) IPN and EDI (Estímulo al Desempeño Académico) fellows and Diana Fischer for English edition.

### Funding

This work was supported by Consejo Nacional de Ciencia y Tecnología (CONACyT), Secretaria de Educación Pública (Project SEP-CONACyT 79707) and Instituto Politécnico Nacional. The funders had no role in study design, data collection and analysis, decision to publish, or preparation of the manuscript.

### Grant Disclosures

The following grant information was disclosed by the authors:
Consejo Nacional de Ciencia y Tecnología.
Secretaria de Educación Pública: SEP-CONACyT 79707.
Instituto Politécnico Nacional.

### Competing Interests

The authors declare that they have no competing interests

### Author Contributions

- Bárbara González-Acosta conceived and designed the experiments, performed the experiments, analyzed the data, prepared figures and/or tables, authored or reviewed drafts of the paper, and approved the final draft.
- Aarón Barraza conceived and designed the experiments, performed the experiments, analyzed the data, prepared figures and/or tables, authored or reviewed drafts of the paper, and approved the final draft.

- César Guadarrama-Analco performed the experiments, prepared figures and/or tables, and approved the final draft.
- Claudia J. Hernández-Guerrero conceived and designed the experiments, authored or reviewed drafts of the paper, and approved the final draft.
- Sergio Francisco Martínez-Díaz conceived and designed the experiments, performed the experiments, authored or reviewed drafts of the paper, and approved the final draft.
- César Salvador Cardona-Félix conceived and designed the experiments, analyzed the data, prepared figures and/or tables, authored or reviewed drafts of the paper, and approved the final draft.
- Ruth Noemí Aguila-Ramírez conceived and designed the experiments, analyzed the data, authored or reviewed drafts of the paper, and approved the final draft.

### Field Study Permissions

The following information was supplied relating to field study approvals (*i.e.*, approving body and any reference numbers):

The scientific collector license was requested from SEMARNAT (Secretaría de Medio Ambiente y Recursos Naturales) (Norma SEMARNAT-08-049b Positiva Ficta).

### Data Availability

The data is available at NCBI SRA: PRJNA760541.

https://www.ncbi.nlm.nih.gov/bioproject/PRJNA760541.

### Supplemental Information

Supplemental information for this article can be found online at http://dx.doi.org/10.7717/peerj.13133#supplemental-information.

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
