# Peer review of "Depth effect on the prokaryotic community assemblage associated with sponges from different rocky reefs"

_PeerJ, doi:10.7717/peerj.13133_

## Round 0.1 · original submission · Major Revisions

We received mixed reviews, from minor to reject. I do not think the reviewers identified any "fatal" flaws, so I encourage you address their concerns and resubmit.

Regards,

Michael

Reviewer 1 ·

Basic reporting

The study by González-Acosta et al describes the impact of depth on bacterial community of marine sponges inhabit different Rocky Reefs. The manuscript specifically focuses on bacterial community of marine sponges in a specific niche that has not much been explored. A study that compares prokaryotic community composition of marine sponges from different depth is rather limited, therefore, this work could provide additional insight on depth impact on sponge-associated bacteria. In general, the structure of manuscript has followed the guidance of the journal and has been well-written. However, there are a number of inaccuracies observed regarding description of methods, presentation of results, the depth of discussions and conclusions which overall reduce the quality of manuscript.

Details are as follow:
Title: The author proposed a title “Depth effect on the bacterial community assemblage associated with sponges from different rocky reefs”. Although from the result indeed the presence of bacterial phyla were more dominant, the authors also reported the small percentage of Archaea. Therefore, the term prokaryotic community would be more accurate instead of bacterial community.

Introduction:
Line 62-63: The author could add 1-2 sentence to explain why they focused to study sponge-prokaryotic community within 2-10 m. Currently there is no explanation in introduction.

Experimental design

Methods:
1. Line 86-87: The comparison of prokaryotic community between Aplysina spp and Ircinia spp is rather inbalance given that Ircinia spp could only be found in 2 m depth and absence in deeper depth (10 m). Why authors did not select other sponge species that present both in 2 and 10 m which would make the comparison and number of samples fairer.

2. Line 90-92: After sponge samples were collected, the authors did not explain how these specimens were identified. After collecting samples, authors must identify the species using COI barcoding or spicules to be sure an exact specimen they were working on. Indeed, sponge species within the same genus could share some prokaryotic phylum, however, at OTUs level such discrepancies can be quite wide. Thus, the comparison must be made between the same sponge species.
Depth is a proxy of other environmental factors; therefore, the additional environmental data is of importance to support the analysis. The authors did not collect water samples from the location of sampling (at least 1 litre in triplicates). The lack of water samples would influence the analysis significantly because there would be lack of an important environmental control regarding prokaryotic community composition. Moreover, the analysis would be more accurate if the authors collected data on light intensity and temperature between the two depth or at least used secondary data from other publication on light pattern and temperature in the sampling location.

3. Line 95: As the standard procedure before extracting DNA, sponge samples must be washed /cleaned with sterile seawater/artificial seawater three times to ensure all debris attached have been removed. In the current manuscript, authors did not explain whether such washing/cleaning was performed. Without a proper cleaning procedure, it is likely that the amplified bacterial communities from sponge specimens have been mixed up with environmental bacteria which could ruin the overall analysis.

4. Line 120-13: The authors did not provide a supplementary file containing R-code and others related documents to perform the community profile analysis. The availability of R-code could be used for readers to reproduce the analysis. The authors also did not mention whether or not the sequence data has been stored in public database such as GenBank or ENA.

Validity of the findings

The authors provided a well-prepared figure to highlight their findings. However, all main figures do not contain title or figure legend which make it challenging to understand the idea of each figure. In addition, the authors could give a more informative result by creating a heatmap that illustrates top OTUs with highest relative abundance among sponge specimens to indicate their contributions within their host. Authors could check papers from Dat et al (https://peerj.com/articles/4970/) or Steinert et al (https://peerj.com/articles/1936/). Since sponge prokaryotic community composition was amplified using the sponge EMP primers, authors could also confirm whether or not the OTU sequences they obtained belong to sponge-enriched OTUs (https://academic.oup.com/gigascience/article/6/10/gix077/4082886) by comparing their result with the dedicated database. The description on how the comparison is performed could also be found in Dat et al.

The term deep that was used to describe sponge samples taken from 10 m is rather inaccurate as the term deep can cover way beyond thousands kilometers. In fact, 2-10 m is still considered as shallow in term of how light penetrates. I would suggest to use the actual depth in the figure legend instead of “shallow” and “deep” to avoid misleading.

Discussion
Although Archaea presence in low relative abundance, their presence also needs to be discussed. At the moment, the authors have mainly focused to discuss the bacterial groups which in general have been well recognized. As mentioned in the result part, the discussion could be deeper if authors analyzed OTUs level and recognize which OTUs within dominant phyla that have changed (increased/decreased) or stable as depth increases.

Conclusion
Line 250-252: The authors concluded that depth plays more important role than host identity to determine prokaryotic community composition. This conclusion could be supported if only data regarding related environmental factors are available and tested. In addition, comparison of sponge bacterial community within sponge specimens that were identified until genus level makes such conclusion weak.

Reviewer 2 ·

Basic reporting

The article is well written and understandable, but the reviewer would recommend rephrasing some sentences for clarity:

Line 54: “which in this last deep water depth”. This is confusing, rephrase please.
Line 65: “The MS of Aplysina species”. Replace for example by Aplysina species, or MS belonging to Aplysina genus.
The last two paragraph in the introduction should be combined and the technical aspects of the sampling, such as sites, depths and so on should be moved to the Materials $ Methods section.
Line 130: Change out by OTU
Line 150-151: “both shallow water samples” is not correct. The authors mean the shallow sponge samples, but how the way the sentence is constructed it is confusing and needs to be change.
Line 155: Remove “(shallow)” as there’s no need to specify as there are only shallow samples for Ircinia. This is something to correct across the whole manuscript.
Line 156: Cyanobacteria was
Line 161: Bacteroidetes was
Line 162:163: rephrase: “Aplysina spp of shallow samples” changed it by “Shallow Aplysina sp. Samples”. Also, spp. refer to several species from the same genus, while sp. Refers to only one. Please check this through the manuscript.
Line 171: “deep water depth samples”: Rephrase. It is something that appears throughout the manuscript, and it is very confusing. It would be easier to use “deep samples” and “shallow samples”.
Line 192: Replace “followed” by “was performed”.
Line 199: “the most abundant phylum was”
Line 203: “This phylum”. Remember that “phyla” indicates the plural of “phylum”.
Line 208: replace “shallower” by “shallow”.
Line 221: they say 40% but in the figure 2 is 30%.
Line 229: Change Fig. 2 by Fig. 3.
Line 246: I would recommend the authors to delete “, and so on”.
Line 248: remove “(shallow and deep)”.

The authors made a good use of the references cited for supporting the research. However, line 203 a reference is missing at the end of the sentence that starts in line 201.
Also, reference Webster et al. 2010 in the “References” section does not appear in the manuscript.

It is recommended that the authors submit the sequences retrieved from this study to a NCBI or similar, to make them available for other researchers. In addition, the accession numbers provided once the submission is completed should be indicated in the manuscript.

Experimental design

In general the experiment is well designed and performed, but essential information must be indicated, as well as some decissions that the authors made before the experiment.
I would recommend the authors to re-organize the last two paragraphs of the Introduction, in order to define a clear aim of the study and avoid repetitions.
Moreover, I encourage the authors to indicate the number of samples that were collected at each depth from each species. This information can be retrieved from the figures but it is necessary to specify it in the “sample collection” part as well.
The authors should specify that what it is in brackets in lines 86 and 87 is the permission that they obtained for SCUBA diving.
The authors said that they isolate the heterotrophic bacteria from the sponges and for doing so they remove the pinacoderm of the sponges. However, this procedure can be a bit arbitrary: how may millimitre s did they remove?; how are they sure that all the phototrophic microorganisms were removed?. On the other hand, should the authors wanted to focus only on heterotrophic bacteria they should point out the reasons why, as in the aims it is not clear.

Validity of the findings

The authors claim that depth model sponge bacterial communities when the results showed that depth cause differences in composition and diversity of microbiome (actually on the heterotrophic bacteria) in one marine sponge species, as for the second sponge studied there were not deep samples.
Furthermore, they bring a step forward and said that microbiomes have the potential to determine environmental perturbations. This statement can actually be true, but the authors need to support this idea with other studies, given that the number of samples and only one species studied at two depth it is not possible to make such a statement.
Therefore, I encourage the authors to move this part of the conclusion to a final paragraph on the discussion

·

Basic reporting

This study examines the microbial community of two species of sponges collected from the western coast of Mexico. There are much fewer of these type of studies from this region than the Caribbean and other parts of the Pacific so it is exciting to see the study.
The text reads well overall but there should be some careful typo and English editing in a few parts for clarity – for example, the last three sentences of the abstract could be clearer and check for typos such as “out” instead of “OTU” (L130).
Also, the primary goal of the study is not clear to me – there are different goals provided at the end of the abstract, L81 at the end of the introduction, and again at the end of the discussion. The implications for climate change seem to come out of nowhere with little context. I think this is relevant, but as written, there is not enough of a connection or background provided. Furthermore, given the experimental design with only one of the two species with ‘deep’ samples, the inferences that can drawn from the dataset are limited and that should be made clear in the writing.

L53-56 unclear and clunky
L62 why is “shallow” in quotes?
L65-67 and 73-77 – these introduce the study species and for Ircinia there is some more context for the interest in the species here, for Aplysina it could use some more context, it seems like a jump from the first sentence of the paragraph to the second sentence. Also, it seems like a sentence or two on the microbial species assemblages (e.g., presence of photosymbionts like Cyanobacteria) is well known for these and would be relevant information here.


Figures-
Fig 3a, what is the vertical line here? Was the OTU table rarefied at this sample size?
Fig5 The strange shape of the ‘deep’ samples in the box plot suggests a larger sample size is needed, this should be mentioned in the discussion/interpretation.
For the supplemental figures, is there a word document with corresponding text?

Experimental design

L92-95 was the whole section of sponge used for DNA extraction or just the pinacoderm, or just the mesohyl?
L133-136 what was used for normalization?
The experimental set up with only one species with ‘deep’ samples makes for limited ability to interpret deep/shallow other than for that one species. This should be made clear in the methods and the text should reflect a more specific inference in the Discussion – maybe using specific language to the species and this dataset would be helpful.
The other studies brought into the discussion particularly 215-239 help make for a more robust discussion, I think the authors could build on this a little more – 236-239 has a start, it would be great to see this expanded by a few sentences.

L246 remove “and so on”.

Validity of the findings

The analysis is sound, although I still want to know about whether there was rarefaction or relative proportions were used or some other normalization approach for the OTUs. And the same note about the lack of context for L251-254.

Additional comments

No further comments.

---

## Round 0.2 · Major Revisions

The revised manuscript received thoughtful suggestions from one reviewer. I also have many comments on style. In short, I suggest revising to write as succinctly and directly. I provide examples for the first 41 lines, which I hope are helpful.

Start with a statement of why sponge microbiomes are important. Do sponges provide ecosystem services, have economic value..?

Line 2. Write actively, directly and succinctly. Replace “Marine sponges are considered harboring one of the richest microbial symbiont communities, which inhabit from shallow to mesophotic ecosystems providing a comfortable place to many symbiotic species.” With “Marine sponges inhabit shallow to mesophotic ecosystems and harbor on diverse symbionts.”

Line 4 -6. This doesn’t follow. How can the community change with depth but also be uniform. Also, avoid phrases like “are also known to.”

Line 6 -7. Notorious has a bad connotation. Also, this is wordy and never comes to stating an unmet need. Abstract should justify the work. Replace “Implementing …these organisms” with “Implementing next-generation sequencing (NGS) has revealed the diversity of the sponge microbiomes varies. The controls of this variation is not well described ..."

Line 12. Replace “to estimate…host” with “to determine the relative importance of environment and host in structuring the microbiome of sponges.”

Line 16. I don’t think you need to identify the core facility or R packages in Abstract.

Line 23. Delete “The analyses showed that” and start with a statement that the most abundant phyla differed between hosts.

Line 26. Don’t repeat methods. Delete “ Through a PERMANOVA analysis, the effect of depth and species on bacterial community structures was estimated”

Line 28. Delete “The” whenever possible. (Also line 31)

Line 29. Replace “Finally, through an unsupervised bi-clustering analysis, outcomes of this approach were found also highlighting clustering of the sponge samples based on depth instead of species.” With “Cluster analysis suggested that depth was more important than host in structuring the sponge microbiome.”

Line 31. Delete “The microbial… (sponge species).”

Line 36. See comment about line 2.

Line 39. Revise “tissues host many symbionts, including…”

Line 41. Replace “Remarkably…associated with marine sponges” with “Marine sponges support between 15 to ## phyla but the source of this variation in diversity is not well described.

Reviewer 2 ·

Basic reporting

The authors have incorporated the first paragraph of the discussion that clarify important aspects related to the number of samples and study design. However, this information should also appear in the introduction and in the methods sections. Otherwise, an original paper in which the main objective is to study the depth effect on prokaryotic communities present in marine sponges, would have planned a deeper sampling design including more species, number of samples, and even more depths if possible. While it is interesting the study showed here, it is important to emphasize these issues.
Furthermore, I recommend the authors to avoid strong statements such as the one in line 32 and 400 (“strongly confirms, suggests”). They have to keep in mind, and write it accordingly, that the results of their research is based in only one sponge species in one location, and thus, this should be indicated in the conclusions.
Moreover, references according to Dat et al., 2018; Erwin et al. 2015; Schmitt et al., 2007; Schmitt et al., 2008; Souza et al., 2019; Steinert et al., 2016; Webster et al., 2013; are missing in the reference list

Experimental design

I disagree with the authors in the reason for what they remove the pinacoderm. If they wanted to remove potential microbial contamination from the seawater they should have submerged the sponge samples in sterile seawater. The authors should know that as filter-feeders the sponges not only are in contact with the microbial communities present in the seawater through the pinacoderm, but through the complex aquiferous system, formed by plenty of channels, they have in order to filter the seawater, what they are doing constantly. In fact, by using this methodology they risk of missing many bacteria present at the sponge periphery. Thus, I recommend the authors to explain in the manuscript the reasons why the removed the pinacoderm from the samples. Whether this procedure was done for the original aim of the study, which was to analyze the bioactive compounds, they should also indicate this in the manuscript. Finally, the interpretation of the results should also be made in accordance to the fraction of sponge tissue analyzed.

Validity of the findings

The authors should consider the original study design while interpreting the results obtained, as a low number of samples and species and only a fraction of the sponge tissue were studied, and they should not generalized the results to other sponge species.

Additional comments

Line 70, specify microbial community.
Line 247 rephrase, “examined” is repeated and lack of sense.
Line 254-255: change “communities prokaryotes” by prokaryotic communities.
Line315-316: link the two sentences.
Line 334-337. The authors say that shallow samples share more OTUs than the samples from the same genus. According to the Fig2b, Aplysina deep and shallow samples share 10.4% OTUs while Aplysina-2m and Ircinia-2m share 6.1%. The authors must to correct this sentence. Moreover, it is worthy to mention that Ircinia samples share 4% of the OTUs with Aplysina-10m, which actually is not that different from the 6.1% between the shallow samples. Furthermore, this contrast to what is said in lines 348-353.
Line 386-389: reference needed.

---

## Round 0.3 · Minor Revisions

We received one review of this resubmission (v2). The other reviewer of v1 did not find it necessary to review this latest again. Accordingly, I am returning the manuscript for minor revisions.

Regards,

Michael

Reviewer 2 ·

Basic reporting

The authors made great changes to the manuscript and clarify an important part of the experimental design. However there are some aspects that need to be change.
Every time that the authors say Aplysina or Ircinia they should write sp. to specify that they mean to a species of such genus.
Line 75: after the comma start a new sentence or add a word such as "and".
Line 103: Replace Goméz by Gómez
Line 122 and 125: replace 16S rDNA by 16S rRNA.
Line 202: the authors mention results for ANOSIM analyses that are not explain in the methods section.
Line 206-207: this sentence should be moved to the discussion
Line 213-220: this large sentence is similar to that in line 75, and needs to be split in two or three for a better understanding. Moreover, replace 16S gen by 16S rRNA gene.
Line 234: remove capital letter from nitrifying.
Line 255: Chloroflexi was
Line 270: reference Pita Galán, 2014 should be Pita, 2014
Line 276-277: merge the two sentences or rephrase.
Line 285: Proteobacteria should be written with capital P.
Line 343-346: I would recommend the authors to remove this sentence. The results they found are interesting but they did not study the effect of environmental perturbations, so that based on their own results they can not confirm this.
Reference Cruz-Barraza & Carballo-Cenizo, 2008 is missing in the reference list
Reference Li et al. 2006 does not appear in the main text
Reference Mohamed et al., 2008 is missing in the reference list
Reference Verdugo Díaz 2004 does not appear in the main text
Reference Webster et al. 2010 does not appear in the main text
Figure 5: include axis labels, and explain to what samples the differences correspond.

Experimental design

no comment

Validity of the findings

no comment

Additional comments

no comment

---

## Round 0.4 · Minor Revisions

We received only one review of the previous version of the manuscript. To move the manuscript forward I reviewed the manuscript and made a number of comments. See attached pdf.

---

## Round 0.5 · Minor Revisions

I think the manuscript is acceptable with some minor edits. Importantly, I think Abstract should have a conclusion. This suggestion and others are indicated below.
Line 22. Revise to
"Conclusions. Microbial diversity analysis showed…microbiome. This observation contrasts with previous reports that the sponge microbiome is highly host specific."
Line 31. Insert commas ( …bacteria, especially…mesophyll, …)
Line 326. Delete phrase “it is worth noting” (let the reader decide)
Line 331. Revise to “in the shallowest samples”
Line 407. Revise to “…revision of…(lowercase)”
Line 466. Cap proper nouns (Caribbean)
Line 482. Delete issue (7:e53029). See also lines 495, 642
Line 499. Only cap proper nouns in titles (Unicellular eukaryotic core…). See also line 513
Line 588. Revise to “Microbial Ecology”
Line 597. Revise to “… Journal 8:1198-…”

---

## Round 0.6 · accepted · Accept

I appreciate your patience through these multiple revisions.

Regards,

Michael

---

## Author Rebuttal · Round 0.6

February 21th, 2022

Dear Editor

I appreciate the corrections made to the manuscript; they have all been incorporated into the document.

Dra. Ruth Noemí Aguila Ramírez
Centro Interdisciplinario de Ciencias Marinas-IPN